# Feasibility and Acceptability of a Telephone-Based Smoking Cessation Intervention for Qatari Residents

**DOI:** 10.3390/ijerph192416509

**Published:** 2022-12-08

**Authors:** Vasiliki Leventakou, Mohammed Al Thani, Angeliki Sofroniou, Hamza I. Butt, Safa M. Eltayeb, Iman A. Hakim, Cynthia Thomson, Uma S. Nair

**Affiliations:** 1Health Research Governance Department, Ministry of Public Health, Doha P.O. Box 42, Qatar; 2Public Health Department, Ministry of Public Health, Doha P.O. Box 42, Qatar; 3Mel and Enid Zuckerman College of Public Health, University of Arizona, Tucson, AZ 85724, USA; 4College of Nursing, University of South Florida, Tampa, FL 33612, USA

**Keywords:** smoking quitline, smoking cessation, Qatar, tobacco control, feasibility

## Abstract

The steady increase in smoking rates has led to a call for wide-reaching and scalable interventions for smoking cessation in Qatar. This study examined the feasibility and acceptability of an evidence-based smoking cessation program delivered by telephone for Qatari residents. A total of 248 participants were recruited through primary care centers and received five weekly scheduled proactive behavioral counseling calls from personnel trained in tobacco cessation and navigation to obtain cessation pharmacotherapy from clinics. Outcomes were assessed at end of treatment (EOT), and 1- and-3-month follow up. The Mann–Whitney test was used to compare the average number of participants recruited per month pre- and post-COVID. We recruited 16 participants/month, the majority (85.5%) attended at least one counselling session, and 95.4% used some of pharmacotherapy. Retention rates were 70% at EOT, 64.4% and 71.7% at 1- and 3-month follow up, respectively; 86% reported being ‘extremely satisfied’ by the program. Our ITT 7-day point prevalence abstinence was 41.6% at EOT, 38.4% and 39.3% at 1-and 3-month, respectively. The average number of participants recruited per month was significantly higher for pre vs. post-COVID (18.9 vs. 10.0, *p*-value = 0.02). Average number of participants retained at EOT per recruitment month showed a slight decrease from 8.6 pre- to 8.2 post-COVID; average number who quit smoking at EOT per recruitment month also showed a decrease from 6 to 4.6. The study results indicated that our telephone-based intervention is feasible and acceptable in this population and presents a new treatment model which can be easily disseminated to a broad population of Qatari smokers.

## 1. Introduction

Cigarette smoking continues to be a global public health problem. In 2019, approximately 1.1 billion people globally used combustible tobacco [1]. According to the World Health Organization (WHO) more than 7 million people die every year because of tobacco use [2] and if tobacco consumption remains unchanged the number of deaths is expected to rise to 8 million per year by 2030 [3]. Smoking rates are particularly high in the Eastern Mediterranean Region (EMR) with more than 30% males currently smoking [4]. According to a recent survey in Qatar, a quarter of the population were tobacco users with more than 20% using some form of combustible tobacco (cigarettes, waterpipe, medwakh and cigar) [5], which is a strong contributing factor to the high prevalence of smoking-related diseases (e.g., ischemic heart disease, lung cancer) in the country [6,7].

This alarming increase in the rates of cigarette use has led public health officials in Qatar to enforce comprehensive tobacco-related policies (e.g., taxation, smoking ban in public places, exposure to second-hand smoking) shown to be effective in reducing tobacco use internationally. Along with other Gulf countries, in 2006 Qatar joined the WHO Framework Convention on Tobacco Control (FCTC) with the primary goal to reduce the negative consequences of tobacco use through evidence-based tobacco cessation services [8]. While Qatar has initiated the implementation of key FCTC measures, evidence-based individualized treatment programs for smoking cessation in the country are distinctly lacking. One such cost-efficient, highly successful, and scalable intervention for tobacco is telephone-based smoking cessation services (e.g., quitlines) [9].

Robust evidence supports the importance of quitlines as a population-level tool to reduce tobacco use [10]. Since the early 1980s, quitlines have been established worldwide in many countries [11] and are programmed to provide interested callers with individualized cessation support with combined pharmacological and evidence-based behavioral support. The success of quitlines relies on their ability to serve as a population-level intervention which is easily accessible, with the potential to expand its services through innovative technologies and digital resources. In the US, smoking quitlines are available in all states and the use of this population-based approach to reducing tobacco for more than 2 decades suggest an effective model that is worth being implemented in other settings [9]. Although most Western countries have implemented smoking quitlines as standard practice for tobacco control, these initiatives are lacking in the EMR countries [8]. Specifically, in Qatar, there is still a paucity of comprehensive behavioral tobacco cessation services with little research for cessation interventions tailored for this population.

As a first step to address this gap, our study examined the feasibility and acceptability (primary outcome), and preliminary results (secondary outcome) of a telephone-based quitline program in Qatar [12,13]. Given that US is the largest tobacco quitline provider, the frequency and structure of the counselling was modelled on US-based quitline programs and intervention content was developed to be culturally responsive to the needs of smokers in Qatar. Next, since the final months of the study recruitment coincided with the outbreak of the COVID-19 pandemic, we also examined the impact of changes to the study protocol and associated changes in recruitment, retention, and quit rates pre- and post-onset of the pandemic.

## 2. Materials and Methods

### 2.1. Study Design and Participants

Briefly, we recruited smoking cessation treatment-seeking male adults, who were residents of Doha. The study recruited only male smokers since it is culturally not widely accepted for women to smoke. Participants were recruited from the smoking clinics of primary health care centers (PHCCs). The smoking clinic physicians examined the patients and prescribed them the appropriate smoking cessation medication (e.g., Champix or Zyban) and/or nicotine-replacement therapy (e.g., nicotine gum, patch, or lozenge). Interested patients were referred to the study staff that was present at the clinic, who provided them a brief overview of the study. Those patients interested in participating in the study proceeded with the eligibility screening. Eligible participants were 18–60 years of age and were daily male smokers (smoking at least one cigarette per day for the past 7 days). Exclusion criteria included active psychosis or non-nicotine drug dependence, exclusive use of electronic cigarettes or Electronic Nicotine Delivery System (ENDS), smokeless tobacco, and/or hookah or shisha use. After completing the informed consent and baseline assessments in-clinic or by phone, a trained smoking cessation counselor followed participants with proactive weekly calls for five weeks. Telephone assessments occurred at end of treatment (EOT), and at one- and three-months post EOT. The study was reviewed and approved by the PHCCs Institutional Review Board (IRB) in Qatar. The study design has been described in detail elsewhere [12].

From October 2020, to comply with social distancing policies caused by the COVID-19 outbreak, information of participants recruited from the PHCCs was sent securely via email to the study staff. Staff then proactively reached out to participants via telephone to seek interest, assess eligibility, obtain informed consent, and assist participants to complete the baseline questionnaires. All modifications were re-reviewed and approved by the PHCC IRB.

Following the study enrollment, participants received five weeks of telephone-based smoking cessation counseling and counselors used strategies to support medication adherence. In brief, the first three sessions focused on increasing motivation to quit, quit preparation, developing coping strategies, setting a quit day, and the final sessions focused on relapse prevention training. Follow-up assessment occurred at EOT (approximately 1 week after Week 5 of counseling) and at one- and three-months post end of treatment. There were no changes to these protocols pre-, and post-pandemic onset. Details of the counseling intervention are presented elsewhere [12].

### 2.2. Measures

Baseline characteristics included demographics, smoking history, and nicotine dependence assessed using the Fagerström test for nicotine dependence (FTND) [14] and were collected at the time of study enrollment. Feasibility outcomes (primary aim) included recruitment rate (number of participants enrolled per month), retention rate (percent retained through the end of treatment and at 1- and 3-month follow-up assessments), compliance to phone sessions, compliance to pharmacotherapy. Acceptability (primary aim) was assessed by participant satisfaction at the 3-month follow-up. Participants were asked how satisfied they were with the quitline program, and they could rank their opinion at a 6-point Likert scale (extremely dissatisfied, dissatisfied, neutral, slightly satisfied, satisfied, extremely satisfied). Self-reported smoking outcomes (secondary aim) were assessed using self-reported 7-day point prevalence abstinence at EOT and one- and three-month follow-up assessments.

### 2.3. Statistical Analysis

Summary statistics (mean ± S.D. or frequency (%) were calculated for demographic and smoking characteristics at baseline. Feasibility and acceptability outcomes were summarized using descriptive statistics. We further stratified outcomes by recruitment timepoint (pre-/post-COVID lockdown). Recruitment rate was calculated as the average number of participants recruited per month during the recruitment period. Since participants’ recruitment was suspended between April and September 2020 due to the COVID-19 lockdown, no recruitment rates were reported for this period. Retention was calculated as the percentage of enrolled participants who completed an assessment at EOT, 1-month or 3-month follow-up; this calculation excluded those that were marked as ‘excluded’ (did not consent to participate or did not meet the inclusion criteria after initial screening) or ‘withdrawn’ (stated they want to withdraw from the study at any point). Compliance with phone sessions was determined using two methods: (a) total number of the phone counselling sessions that the participants attended (maximum 5 sessions); (b) the attendance status for each of the 5 phone sessions (yes/no). Participants self-reported usage of pharmacotherapy was assessed at 3-month follow up. Finally, we calculated smoking quit rates as point-prevalence abstinence from smoking during the last 7 days, or 30 days, respectively (based on smoking even a single cigarette in the last 7 or 30 days) according to self-reported responses at EOT, 1-month follow-up, and 3-month follow-up assessments. The Mann–Whitney test was used to compare the average number of participants recruited per month pre- and post-COVID. All statistical analyses were carried out using SAS 9.4 (SAS Institute, Cary, NC, USA) and Microsoft Excel.

## 3. Results

### 3.1. Characteristics

Overall, n = 676 smokers were referred to the quitline and n = 296 were eligible for the study (Figure 1). Of the enrolled (n = 248) smokers, 212 (85.5%) attended at least one counselling session and 155 (63%) completed the EOT.

Characteristics of the participants are summarized in Table 1. Participants had a mean age of 38.5 ± 9.0 years, the majority were Egyptians (29.8%), married (82%), had a college degree (66.5%) and were employed (93.5%). The majority were daily smokers and smoked an average of 16.9 (±11) cigarettes per day. Almost half of the participants (45.3%) reported a complete home smoking ban, 30.5% had some bans, and a quarter of the sample reported that smoking was allowed anywhere in the home.

Pre-COVID participants were on average slightly older (39.1 ± 9.1 years) than post-COVID participants (35.8 ± 8.4 years). Majority pre-COVID were Egyptian (33.0%) whilst majority post-COVID were Jordanian (26.7%). There was also a larger proportion of currently married participants pre (83.5%) vs. post-COVID (75.6%). Mean FTND score was slightly higher in pre- (4.5 ± 1.0) vs. post-COVID (4.2 ± 1.0). House smoking rules showed a similar pattern between the two groups.

### 3.2. Primary Outcomes

#### 3.2.1. Feasibility

We recruited approximately 16 participants per month. A majority (86%) of enrolled participants received at least one counselling session. The median number of phone sessions attended was two and the average time for each counselling call was 13.3 (SD) minutes. Participants could be prescribed with more than one of the available medications by their physician. Approximately 87% reported some use of Champix, 67.8% reported using the nicotine lozenge, 49.3% reported using the nicotine patches, and only 2.6% used gums. End of treatment retention rates was 70% (n = 153), 64.4% (n = 141) for 1 month follow up, and 71.7% (n = 157) for 3-month follow-up time points, respectively.

#### 3.2.2. Acceptability

During the 3-month follow up, the participants were able to rate how satisfied they were by the program selecting from 1 (extremely dissatisfied) to 6 (extremely satisfied). More specifically, the majority (85.9%) reported being ‘extremely satisfied’ and 13.1% were ‘satisfied’. The respondents could rank their opinion at a 6-point Likert scale (‘extremely dissatisfied’, ‘dissatisfied’, ‘neutral’, ‘slightly satisfied’, ‘satisfied’, ‘extremely satisfied’).

#### 3.2.3. Quit Rates

Our self-reported 7-day point prevalence quit rates were 59.5% at end of treatment, 59.6% at one month, and 56.7% at 3 months follow-up. Intention to treat analysis showed lower rates at the follow up timepoints (41.6% at EOT, 38.4% and 39.3% at 1 and 3-months) as shown in Table 2.

### 3.3. Pre- and Post-Pandemic Onset Comparison

The primary and secondary outcomes differed somewhat pre- and post- pandemic. The average number of participants recruited per month was significantly higher for pre vs. post-COVID (18.9 vs. 10.0, Mann–Whitney test *p*-value= 0.02). A slight increase was seen only for pharmacotherapy use and compliance to phone counselling sessions especially after the third session. Average number of participants retained at EOT per recruitment month showed a slight decrease from 8.6 pre- to 8.2 post-COVID; similarly, average number who quit smoking at EOT per recruitment month also showed a decrease from 6 pre- to 4.6 post-COVID (Figure 2). The EOT retention rate (Table 2) was significantly lower for pre vs. post-COVID (0.65 vs. 0.87, *p*-value= 0.003), whilst the 7-day point prevalence smoking quit rate at EOT was significantly higher for pre vs. post-COVID (0.65 vs. 0.44, *p*-value= 0.02). There was no significant correlation between 7-day point prevalence smoking quit status and baseline FTND score in pre- (r = −0.06, *p*-value = 0.735) and post-COVID (r = −0.05, *p*-value = 0.627).

## 4. Discussion

The goal of our study was to assess the feasibility of a telephone-based smoking cessation program, which was integrated within the infrastructure of primary healthcare providers in Qatar. Our results indicate that our intervention is feasible in this population of smokers and participants (of those who completed the program) reported high levels of program satisfaction. While not powered for efficacy, our preliminary evidence indicates that our 7-day point prevalence quit rates at end of treatment and follow-ups exceeded those in other non-western tobacco cessation programs [15,16,17].

In a previous survey of current smokers in Qatar, [18], 67% had expressed willingness to quit; however, the high prevalence of smokers highlights the need to bolster support for smoking cessation in the country [19,20]. In a prospective randomized control trial, El Hajj et al. compared the effectiveness of a face-to-face structured smoking cessation program conducted by trained pharmacists [21]. Our telephone-based cessation program was a feasibility, acceptability, and preliminary efficacy study conducted by trained behavioral counselors. Thus, while the outcome for both of studies was smoking cessation, the study protocols, including the follow-up assessment timepoints varied significantly. The study by El Hajj et al. included adult smokers in Qatar, and it was observed that self-reported 7-day point prevalence quit rates were higher in the intervention group (n = 167, 30.7%) compared to the control (n = 147, 26.5%) at 3-months follow up. While our study did not include a control group, our intervention model achieved quit rates of 56% (ITT rate of 40%)—higher than that of an in-person intervention. Apart from retention rates at 3 months (63% and 71% for the El Hajj et al. and our study, respectively), and the quit rates, there were no data on recruitment rates and acceptability data available for us to allow a further comparison on results. Moreover, our smoking quitline is highly scalable, disseminable, easily accessible, and convenient which eliminates potential barriers to healthcare (e.g., travel, clinic operational time, lack of time). A larger and fully powered study is needed to test the effectiveness of our intervention; however, the preliminary results propose a model that can be adopted by healthcare agencies and thereby have a positive impact on tobacco rates in the Middle East, a region with a high prevalence of tobacco use [5].

The primary healthcare centers (PHCCs) in Qatar serve all residents and as of now 7 out of total 28 available centers have a dedicated smoking clinic with a trained smoking cessation physician and nurse. Residents whose PHCC does not have a smoking clinic are referred to the closest and most convenient clinic for treatment. Our pilot feasibility study suggests the incorporation of the telephone-based design into the existing PHCC clinics which was successful and well-accepted, as shown by the high rate of satisfaction from the participants, is a viable opportunity to reduce tobacco use in the population. Additionally, this approach, which applies a combination of pharmacotherapy and behavioral therapy for smoking cessation is considered the gold standard in healthcare has been shown to be cost-effective [22].

The disruption of the program by the COVID-19 pandemic allowed us to observe the effects of two different referral systems: active (counselors recruited participants on-site at the clinics) and passive (counselors received contact details of interested patients from physicians). Even though the time of recruitment for the active referral was shorter than the passive, the results showed comparable feasibility for both enrolment procedures, similar to other studies [11]. We did see a higher retention at EOT post-COVID compared to pre-COVID. This observation could be explained by the societal changes due to the pandemic (e.g., loss of employment, work remotely, experiencing stress and boredom). Our participants may have had more time to receive the counseling or were more willing to talk or share their thoughts because of their home privacy.

The study has a few limitations that should be acknowledged. In our study we did not bioverify the quit outcomes. The self-reporting of quit rates may have led to an overestimation of the abstinence rates [23,24]. With respect to the social/cultural norms of the country we enrolled only male participants which may have introduced selection bias. A more thorough investigation of the proposed tailored intervention should include bioverified quit outcomes, a larger sample size, and be expanded to women as well. Finally, we are aware that our study was a single group pre-post design which limited our ability to compare results to a true control group. However, in the distinct absence of telephone-based tobacco cessation programs in Qatar, ours is the first study to implement such an evidence-based scalable practice. Since the first step prior to conducting a randomized trial is to test feasibility and preliminary efficacy and in the absence of available data in the literature for this specific population, we chose a single group pre- and post-test design. The main goal was to develop the infrastructure for a smoking cessation quitline in Qatar, a country with specific population characteristics, cultural practices, and with free to low-cost medication coverage for all residents. This study was integrated within the country’s health care system and adapted the existing evidence-based cessation counselling of the Arizona quitline [12,13]. Thus, although the single group significantly limits generalizability, the results are encouraging given the high dissemination potential of our approach. The next step for our study team in this line of research is to develop more rigorous study design to determine efficacy of our experimental design to a standard care/attention control group.

## 5. Conclusions

Our study demonstrated that a telephone-based smoking cessation intervention for smokers who were residents of Qatar is feasible and acceptable and preliminary evidence showed high self-reported quit rates that were sustained through the 3-month follow-up. Given the scalability and dissemination potential of our approach to widely reach a broad population of smokers, the next steps call for a larger scale fully powered randomized control trial to test the effectiveness of our intervention on long-term quit outcomes for this population. Our promising findings also suggest potential for remote treatment in primary health care facilities, especially after the experience of COVID-19 and the shift to remote care.

## Figures and Tables

**Figure 1 ijerph-19-16509-f001:**
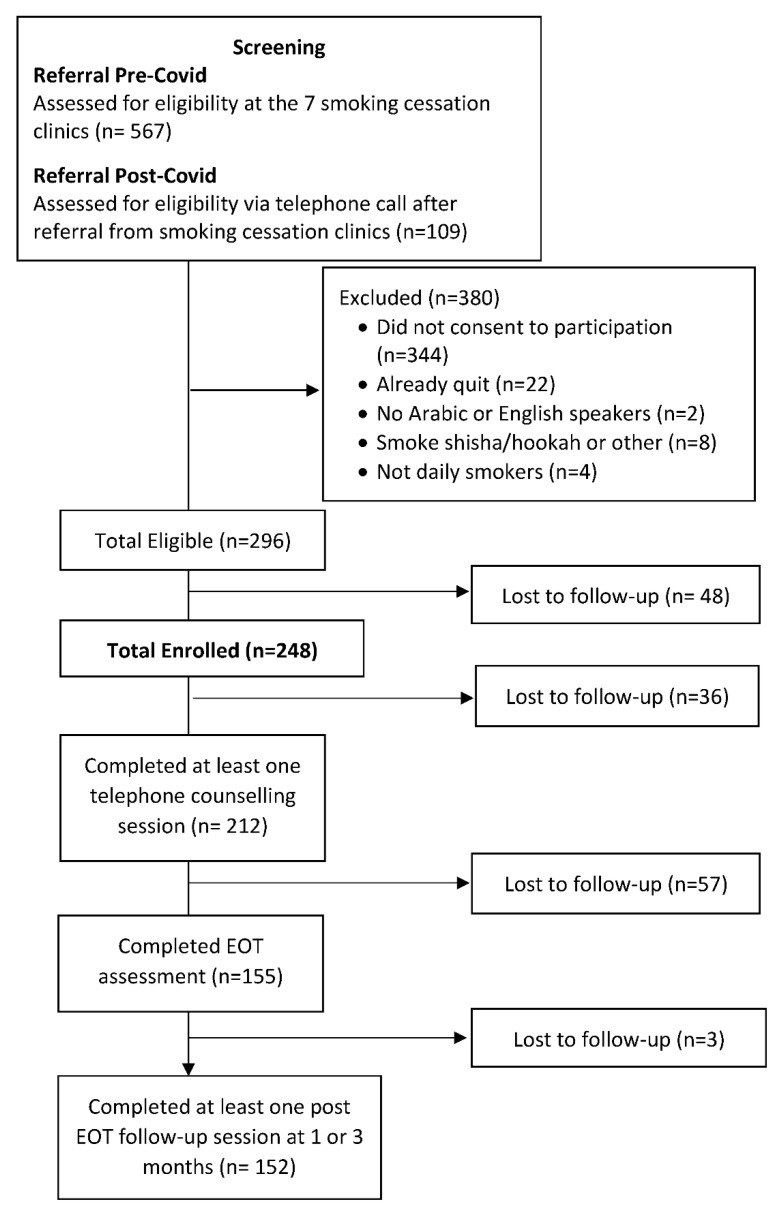
Study population screening and enrollment flow.

**Figure 2 ijerph-19-16509-f002:**
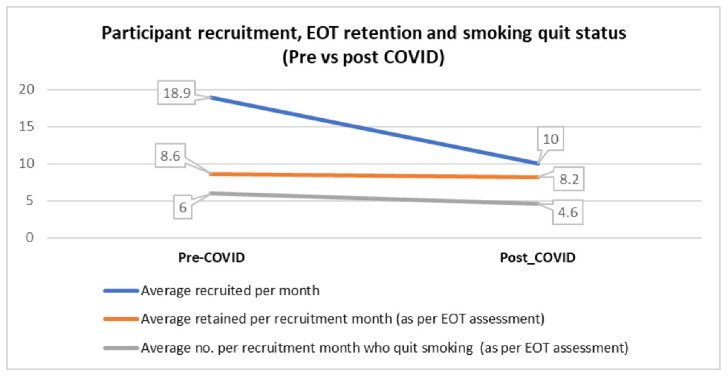
Pre- and post-COVID trends for recruitment, retention, and smoking quit rates.

**Table 1 ijerph-19-16509-t001:** Baseline characteristics of participants, overall (n = 248) and pre- (n = 203) and post-COVID (n = 45) pandemic.

Characteristics	Total (n = 248)	Pre-COVID (n = 203)	Post-COVID (n = 45)
Age (years), mean ± SD	38.5 ± 9.0	39.1 ± 9.1	35.8 ± 8.4
Nationality, n (%)			
Qatari	31 (12.5)	27 (13.3)	4 (8.9)
Egyptian	74 (29.8)	67 (33.0)	7 (15.6)
Jordanian	34 (13.7)	22 (10.8)	12 (26.7)
Syrian	23 (9.3)	12 (5.9)	11 (24.4)
Other ^a^	63 (25.4)	53 (26.1)	10 (22.2)
Missing	23 (9.3)	22 (10.8)	1 (2.2)
Marital Status ^b^, n (%)			
Married or living with partner	201 (82.0)	167 (83.5)	34 (75.6)
Currently not married or living with partner	44 (18.0)	33 (16.5)	11 (24.4)
Education level ^b^, n (%)			
Less than college degree	83 (33.5)	68 (33.5)	15 (33.3)
College degree and above	165 (66.5)	135(66.5)	30 (66.7)
Employment status ^b^, n (%)			
Employed	231 (93.5)	192 (95.1)	39 (86.7)
Not employed	16 (6.5)	10 (5.0)	6 (13.3)
Number of years of regular cigarette smoking, mean ± SD	18.4 ± 8.4	18.8 ± 8.7	16.7 ± 6.7
Number of daily smoked cigarettes, mean ± SD	16.9 ± 11.0	17.7 ± 11.0	13.0 ± 10.1
Nicotine dependence (FTND score), mean ± SD Number of smoking quit attempts over last year, n (%)	4.4 ± 1.0	4.5 ± 1.0	4.2 ± 1.0
None	150 (61.7)	120 (60.6)	30 (66.7)
1	56 (23.1)	48 (24.2)	8 (17.8)
≥2	37 (15.2)	30 (15.2)	7 (15.6)
E-cigarette use in the past 30 days ^c^, n (%)			
Yes	45 (18.2)	43 (21.3)	2 (4.4)
No	197 (79.8)	154 (76.2)	43 (95.6)
Smokers in household beside yourself ^b^, n (%)			
Yes	52 (21.1)	44 (21.8)	8 (17.8)
No	195 (78.9)	158 (78.2)	37 (82.2)
House smoking rules ^c^, n (%)			
No ban	59 (24.3)	46 (23.2)	13 (28.9)
Some ban	74 (30.5)	60 (30.3)	14 (31.1)
Complete ban	110 (45.3)	92 (46.5)	18 (40.0)

Abbreviations: FTND, Fagerström score of nicotine dependence. ^a^. ‘Other’: group of nationalities (Algerian, Bangladeshi, Canadian, Greek, Indian, Iranian, Iraqi, Lebanese, Moroccan, Nigerian, Pakistani, Palestinian, Philippino, Romanian, Saudi, Sudanese, Tunisian, Yemeni) with less than n = 15 participants each. ^b^. Individual (n = 1) responded ‘Don’t know’/‘Refused to answer’. ^c^. Individuals (n = 5) responded ‘Don’t know’/‘Refused to answer’.

**Table 2 ijerph-19-16509-t002:** Feasibility, acceptability, and quit outcomes (overall and stratified by pre-, post-COVID pandemic).

	Pre-COVID (n = 246)	Post-COVID (n = 50)	Overall (n = 296)
Recruitment rate (Avg recruited/month)	18.9	10.0	16.4
Retention ^a^, n (%)			
End of treatment	112 (65.1)	41 (87.2)	153 (69.9)
1-month follow-up	100 (58.1)	41 (87.2)	141 (64.4)
3-month follow-up	113 (65.7)	44 (93.6)	157 (71.7)
No. of phone counselling sessions attended ^a^, n (%)			
0	29 (16.9)	2 (4.3)	31 (14.2)
1	29 (16.9)	5 (10.6)	34 (15.5)
2	36 (20.9)	5 (10.6)	41 (18.7)
3	38 (22.1)	13 (27.7)	51 (23.3)
4	28 (16.3)	17 (36.2)	45 (20.6)
5	12 (7.0)	5 (10.6)	17 (7.8)
Use of smoking cessation medication ^b^, n (%)			
Champix or Chantix	98 (88.3)	34 (82.9)	132 (86.8)
Zyban (Buproprion, Wellbutrin)	1 (0.9)	1 (2.4)	2 (1.3)
Gum	3 (2.7)	1 (2.4)	4 (2.6)
Patch	50 (45.1)	25 (61.0)	75 (49.3)
Lozenges	71 (64.0)	32 (78.1)	103 (67.8)
At least one type of medication	109 (98.2)	36 (87.8)	145 (95.4)
7-day point prevalence smoking abstinence ^b^, n (%)			
End of treatment	73 (65.2)	18 (43.9)	91 (59.5)
1-month follow-up	65 (65.0)	19 (46.3)	84 (59.6)
3-month follow-up	68 (61.3)	18 (43.9)	86 (56.7)
7-day point prevalence smoking abstinence, intent to treat analysis ^c^, n (%)			
End of treatment	73 (42.4)	18 (38.3)	91 (41.6)
1-month follow-up	65 (37.8)	19 (40.4)	84 (38.4)
3-month follow-up	68 (39.5)	18 (38.3)	86 (39.3)
30-day point prevalence smoking abstinence ^b^, n (%)			
End of treatment	78 (69.6)	23 (56.1)	101 (66.0)
1-month follow-up	67 (67.0)	21 (51.2)	88 (62.4)
3-month follow-up	71 (64.0)	20 (48.8)	91 (60.0)
30-day point prevalence smoking abstinence, intent to treat analysis ^c^, n (%)			
End of treatment	78 (45.4)	23 (48.9)	101 (46.1)
1-month follow-up	67 (40.0)	21 (44.7)	88 (40.2)
3-month follow-up	71 (41.3)	20 (42.6)	91 (41.6)
Participant satisfaction at 3-month follow-up ^d^, n (%)			
Neutral	1 (1.7)	0 (0.0)	1 (1.0)
Satisfied	12 (20.7)	1 (2.4)	13 (13.1)
Extremely satisfied	45 (77.6)	40 (97.6)	85 (85.9)

^a^. Calculated based on participants that were not marked as ‘excluded’ or ‘withdrawn’. n = 219 (overall), n = 172 (pre-COVID), n = 47 (post-COVID). ^b^. Calculated for participants that had an EOT, 1- or 3-month follow-up assessment available. Smokers could be single or multi-users of the available medication and thus % does not add up to 100. ^c^. Calculated based on participants that were not marked as ‘excluded’ or ‘withdrawn’. n = 219 (overall), n = 172 (pre-COVID), n = 47 (post-COVID). Participants with a missing assessment at EOT, 1- or 3-month follow-up are assumed to be continued smokers. ^d^. Calculated for participants that had a 3-month follow-up assessment available (n = 99). Participants who responded ‘Don’t know’ were also excluded from the calculation.

## Data Availability

The dataset analyzed during this study is available from the corresponding author upon reasonable request.

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
