# Peer review of "Feasibility and Acceptability of a Telephone-Based Smoking Cessation Intervention for Qatari Residents"

_ijerph, 2022, doi:10.3390/ijerph192416509_

Round 1
Reviewer 1 Report
Suggestions for authors are attached below.

Author Response
Dear reviewer,
We would like to thank you for your meaningful comments to improve our manuscript. We have now modified the paper following these comments. You may find our point-by-point answers below:
Reviewer 1:
"Tobacco consumption remains one of the greatest threats to public health. There are some considerations regarding this study that I would like to comment:"
Introduction:
Comment 1: I miss an explanation for using an US based quit line model, to the Qatar population, and refer it in the test.
Our response: As suggested by the reviewer, we have now added a text to support why we selected a US-based quit line model in our study (page 2, lines 19-21 and line 28).
Methods:
Comment 2: The most important methodological limitation of the study is the lack of a control group during the intervention.
Our response: We thank the reviewer for this comment, and we have further elaborated on this limitation in the discussion section (page 10, lines 31-46). Specifically, there is a distinct absence of telephone-based tobacco cessation programs in Qatar, despite their resounding success in western countries. Ours is the first study to implement such an evidence-based scalable practice and has potential for wide reaching benefits to a high-risk group of individuals who smoke. Secondly, since the first step prior to conducting a randomized trial was to test feasibility and preliminary efficacy of such an approach and in the absence of available data in the literature for this specific population, we chose a single group pre-post- test design. The primary goal was to develop the infrastructure for a smoking cessation quitline in Qatar, a country with specific population characteristics, cultural practices, and with free to low-cost medication coverage for all residents. Although the single group significantly limits generalizability (as we have pointed out in the limitations), the results are encouraging given the high dissemination potential of our approach. Thus, we believe that the results and benefits of our study outweigh the study design methodological limitations. The next step for our study team in this line of research is to develop more rigorous study design to determine efficacy of our experimental design to a standard care/attention control group.
Comment 3: It would be interesting to explain how many calls attempts the counselor did in order to interview the patient and classified the call. Is there any patient already under treatment during the first 5 intervention calls?
Our response: Approximately 24-48 hours post-baseline the assigned quit counselor called the patient to start the intervention. In most cases participants replied to that first call as they had agreed to participate in the study. In general, the counselors made up to 10 call attempts at different time and days to reach participants at any time point of the intervention. If no reply, participants were classified as ‘lost to follow-up’.
Pharmacotherapy use was assessed at end of treatment (EOT), 1- and 3- months follow-up time points and reflects the use during all five intervention sessions. Unfortunately, we have not recorded the pharmacotherapy use at each intervention call.
Comment 4: Is there any patient that started smoke cessation without taking any medication?
Our response: Yes, we had seven participants that reported use of no medication during the smoking cessation process. Out of these seven participants, two reported a 7-day point prevalence abstinence and one participant reported 30-day point prevalence abstinence at the 3-month follow-up.
Comment 5: From the initial sample of 567 participants, 380 were excluded and, from those, 344 refused to participate in the study. The lack of acceptance may affect the sample representativeness?
Our response: We thank the reviewer for this comment. We agree that response rate is a main concern in epidemiological studies, and it is a bias that increases the risk of generalizability. The purpose of the current feasibility study was to identify all potential biases and test readiness of the intervention when scaling to a larger size trial. The next step for our study team is to improve the study design and increase response rates for our population for further testing in a full-scale trial.
Comment 6: There are important differences between the size of the samples before and after Covid (pre: 246, post: 50). Could this influence your results?
Our response: We thank the reviewer for this comment. It may be possible that the sample size may have impacted our results but exploring this is beyond the scope of this study.
Comment 7: Why do the authors consider just a single call sufficient for patient’s follow-up in the study if most them received between 2 and 3 calls (as is shown in Table 2).
Our response: Table 2 reflects the number of phone counselling sessions that participants received. Follow-up assessment occurred at EOT (approximately 1 week after the last counselling call) and at 1- and 3-months post end of treatment. We made up to 10 call attempts to reach out to participants for their follow-up.
Comment 8: It could influence the cessation rate that the mean of FTND score were 4, which means a low- moderate nicotine dependence? Are the quit rates influenced by the level of tobacco dependence?
Our response: Thank you for your comment. We further analyzed quit rates by nicotine dependence and no significant correlation was found between 7-day point prevalence smoking quit status and baseline FTND score in pre- (r=-0.06, p-value=0.735) and post-Covid (r=-0.05, p-value=0.627). (page 8, lines 19-21)
Comment 9: There are some references that appear in the text without explanation, that are commented later on the test: The study design has been described in detail elsewhere [12]. Briefly, we recruited....it could be interesting to change the order to facilitate the reading.
Our response: We have now made the suggested changes by the reviewer (page 3, lines 2-3 and 17-18).
Results:
Comment 10: Table 1: it will be interesting to compare the characteristics of the sample pre and post Covid.
Our response: As suggested by the reviewer we have now updated Table 1 (page 5) including the characteristics of our study population pre- and post-Covid. For this, we also added a description in the results section (page 5, lines 8-13).
Comment 11: Line 1, 2, 3: It is not clear this sentence: Approximately 87% reported some use of Champix, 67.8% reported using the nicotine lozenge, 49.3% reported using the nicotine patches, and only 2.6% used gums. It seems that patients used different medications at the same time (Champix, Nicotine patches) during the follow up. Please, explain it.
Our response: We thank you for your comment. In many cases physicians prescribed combination of medications and patients could be single or multi-users of available medications according to their needs. This is also mentioned as footnote (b) in Table 2 to explain why the % for the ‘use of smoking cessation medication’ does not add up to 100. To clarify this, we have now added in the text the following: ‘Participants could be prescribed with more than one of the available medications by their physician’ (page 6, lines 12-13).
Comment 12: Are there any differences in the retention rates due to the different drugs using for the patients? Champix, nicotine patches...
Our response: As most patients were multi users of the available medications we were not able to investigate retention rates by pharmacotherapy use.
Discussion:
Comment 13: Line 12. We did see higher retention at EOT post COVID compared to pre COVID. Do you have and explanation for this result?
Our response: We hypothesize that the societal changes due to the pandemic (e.g., loss of employment, work remotely, experiencing stress and boredom) may have led to participants having more time to receive the counselling or were more willing to talk or share their thoughts because of their home privacy (page 10, lines 14-18). We feel that while a more in-depth investigation of study milestones and participant demographics will be informative, this is beyond the scope of this paper.
Comment 14: Line 19: a randomized trial, El Hajj et al. examined efficacy of an intensive in-person smoking cessation program.... and it was observed that self-reported 7-day point prevalence quit rates were higher in the intervention group (n=167, 30.7%) compared to the control group (n=147, 26.5%) at 3-months follow up. While our study did not include a control group, 24 our intervention model achieved quit rates of 56%.
Could you discuss the results of your study in comparison with the one that you mention before?
Our response: We have now elaborated more in the discussion section. Please check the following (page 9, lines 13-20 and lines 25-28): ‘‘In a prospective randomized control trial, El Hajj et al. compared the effectiveness of a face-to-face structured smoking cessation program conducted by trained pharmacists [21]. Our telephone-based cessation program was a feasibility, acceptability, and preliminary efficacy study conducted by trained behavioral counselors. Thus, while the outcome for both of studies was smoking cessation, the study protocols, including the follow-up assessment timepoints varied significantly. The study by El Hajj et al. included adult smokers in Qatar, and it was observed that self-reported 7-day point prevalence quit rates were higher in the intervention group (n=167, 30.7%) compared to the control (n=147, 26.5%) at 3-months follow up. While our study did not include a control group, our intervention model achieved quit rates of 56% (ITT rate of 40%)- higher than that of an in-person intervention. Apart from retention rates at 3 months (63% and 71% for the El Hajj et al. and our study respectively), and the quit rates, there was no data on recruitment rates and acceptability data available for us to allow a further comparison on results.’’
Limitations:
Comment 15: It could be a selection bias related to the sample due to their recruitment from Primary Care centers. Are these patients more aware from smoking cessation? Are at the same heath conditions than the general smoking population?
Our response: As per the country’s health care system, all residents are registered at a local primary health care clinic (PHCC) to receive fully integrated healthcare (more information is included in reference 12). Currently, there are 27 PHCCs in Doha and at least 7 have integrated smoking cessation clinics. Smokers who are interested in acquiring cessation assistance are referred to these smoking clinics. The medication is also provided by the pharmacies established within each health care and the cost is fully covered for nationals and 75% covered for residents by the national healthcare system. The goal of our study was to implement a referral system within the existing infrastructure of the PHHCs to refer potential smokers to our telephone-based behavioral counseling program. Thus, there is no selection bias due the location of recruitment.
Reviewer 2 Report
Overall, this paper is very interesting and covers a critically important topic. The study is well-done and analyses are appropriate. There are a few places that need proofreading and formatting (e.g., different fonts in Figure 1, Pre-COVID and Post_COVID in Figure 2). The discussion of the impacts of COVID are very helpful and well-done. Overall this is really important work and with a few edits/attention to detail with formatting and proofreading the paper will be ready for publication.
Author Response
Dear reviewer,
We would like to thank you for your comments to improve our manuscript. We have now modified the paper following these comments.
Reviewer 2:
‘‘Overall, this paper is very interesting and covers a critically important topic. The study is well-done and analyses are appropriate. There are a few places that need proofreading and formatting (e.g., different fonts in Figure 1, Pre-COVID and Post_COVID in Figure 2). The discussion of the impacts of COVID are very helpful and well-done. Overall this is really important work and with a few edits/attention to detail with formatting and proofreading the paper will be ready for publication.’’
Our response: We have now updated the manuscript and changed the fonts of the figures.
Round 2
Reviewer 1 Report
The authors have carried out the suggested modifications. We cannot be certain that the results are due to the intervention, since we do not have the necessary control group to confirm them. This would be the major limitation of the study and the next approach.